# Cohort profile: The Dutch wound monitor cohort and the Swedish Region Halland Integrated Platform (RHIP) wound cohort

Oskar Gustafsson[1,2], Jens Lundström[1], Mattias Ohlsson[1,3], Hanna Stenhamre[2], Daniel Tsang[2], John Pavia[2], Ernst Ahlberg[2,4]

**1** The Department of Intelligent Systems and Digital Design, Halmstad University, Halmstad, Sweden, **2** Clinical Affairs, Mölnlycke Health Care, Gothenburg, Sweden, **3** Centre for Environmental and Climate Science, Lund University, Lund, Sweden, **4** Centre for Reliable Machine Learning, Royal Holloway, University of London, London, Great Britain

## Abstract

Hard-to-heal wounds are a growing human and financial concern, constituting approximately 1–3% of the healthcare budget. Wound care is not a medical specialty and is often not prioritized within healthcare. A large portion of the cost and suffering caused by wounds has the potential to be mediated through improved knowledge and effectivised workflows. One potential way to achieve this is through the implementation of AI-tools to support clinicians in planning and executing wound care. Information-driven care is a framework for implementing AI-technology in healthcare. Wound Monitor is a Dutch database containing data collected from home-care visits conducted by wound specialists during 2005 to 2022, mostly in Limburg. It contains data of more than 17000 patients. Region Halland, Sweden, created a platform of integrated clinical, financial and operational data called "The Regional Healthcare Information Platform" (RHIP). The platform contains data on over 500 000 patients during 2008–2021. Within this data, a subset of almost 39000 patients have been diagnosed with wounds or wound related conditions. This subset of patients are defined as the RHIP Wound Cohort. This article characterizes the two wound cohorts in terms of demographics and wound types. Further, it examines the quality, quantity and granularity of the respective databases. The discussion section evaluates the strengths and weaknesses of the datasets in terms of the perspective they provide on the patient and wound journey. Lastly, the discussion section also explores how the cohorts may be utilized for predictive modeling and other machine learning-based applications in order to enable information-driven wound care.

## Introduction

Hard-to-heal wounds, usually defined as wounds not healing within 4-14 weeks [1], causes pain, suffering and reduces quality of life of patients. Wound care is expensive, it is estimated that 1-3 % of expenses in healthcare is spent on wound care

**Data availability statement:** The data cannot be shared publicly as it contains sensitive patient information. To discuss collaborations involving the Wound Monitor Database, contact the Clinical Affairs department at Mölnlycke Health Care at clinicalresearch@molnlycke.com For the RHIP database, data access requests may be directed to: • Region Halland via email at datauttag.forskning@regionhalland.se • Or to the School of Information Technology (ITE) at Halmstad University via email at registrator@hh.se or phone at +46 35 16 71 00. Details can be found on the ITE website. All requests will be handled in accordance with applicable regulations and institutional policies to ensure long-term data availability.

**Funding:** This work was conducted as part of the CAISR Health Research Profile at Halmstad University with funding made possible by the Knowledge Foundation (grant number 20200208 01H). The funders had no role in study design, data collection and analysis, decision to publish.

**Competing interests:** The authors have declared that no competing interests exist.

[2]. Wound care is not established as a separate specialty for medical professionals in Sweden and standard of care varies substantially across different geographical regions [3]. Complications due to lack of knowledge, inefficiency, poor compliance with guidelines and a lack of belief that clinicians actions can make a difference contributes to a large extent to the costs [3]. There has been much improvement in products and treatments for chronic wounds such as advanced dressing and growth factors. Despite this, much of real world practice does not make use of these innovations and evidence-based treatment [1,3,4]. With an aging population, chronic wounds is expected to become more prevalent [3].

*Information-driven care* is a framework for utilizing artificial intelligence in health-care. The framework describes a workflow starting with the treatment of patients being documented, generating data. This data is being analyzed to gain insights on how to implement changes that will improve care. The implemented changes will be observed in new data which can be analyzed again [5]. This iterative process should be thought of as a continuous and never-ending process. The rapid developments in machine learning and AI opens up new possibilities for identifying complex patterns in real-world data.

This article is a profile of two real-world cohorts that enables two different and important perspectives on wound care. The materials and methods section outlines the data-collection process and exclusion and inclusion criteria. The results section outlines the characteristics of the datasets. The discussion section outlines the possibilities that these real-world datasets open up for *information-driven wound care*. This includes evaluating their strengths and weaknesses in terms of enabling a patient- and wound journey analysis, the generalizability of potential findings and possibilities for developing machine-learning based applications.

To the best of our knowledge, there are no other cohort profiles in wound care. However, there are cohort studies in wound care which are similar to cohort profiles in some regards. Guest et al. [6] analyzed a cohort of patients (n = 3000) with wounds from a database with patients medical history. They investigated the health economic burden of wounds in Great Britain from retrospectively analyzing electronic health records. Lo et al. [7] studied the clinical and economic burden of wounds in the Tropics. Carville et al. [8] analyzed the costs and healing outcomes of three different Australian wound cohorts. There are also, to the best of our knowledge, no studies that focuses on how multiple cohorts in wound care can complement each other in ongoing work. There are wound registries all over the world but since no cohort profile has been published and the data is private, a direct comparison with these in terms of data capture and granular demographics is not possible at the time being. When comparisons are possible, for example with high-level demographics such as age and gender, differences and similarities are systematically evaluated.

## Datasets

The Wound Monitor database is data collected from Dutch home-care clinics during 2005–2022. The majority of patients are located in Limburg. A substantial minority are located in North Brabant and a few patients are located in other parts

of the Netherlands. The database is visit- and wound-centered and contains wound assessments, wound outcomes, demographic and clinical variables, wound images, products used in wound treatments and treatment plans. Wound assessments occur regularly, often weekly. At each assessment, a standardized set of variables are collected.

The Regional Healthcare Information Platform (RHIP) in Halland, Sweden is a platform of clinical and administrative records from all consumers of publicly funded healthcare since 2009 in Region Halland (RH). The platform is visit- and patient-centered and contains clinical, financial and operational capacity data from 77 primary care units and three hospitals encompassing more than 500 000 patients. The data is structured in a relational database. The database tables include information such as patient demographics, diagnosis, medical and procedural treatments, lab-values, vital parameters, cost of treatments and professions of the medical providers [9]. For patients with wounds, the majority of visits occurred in primary care units while a substantial minority of visits occurred at the emergency departments and hospitals. A small minority of visits were performed in specialized skin care units.

### Inclusion and exclusion criteria

The wound monitor dataset contains data on 18845 patients. However, only 17547 are linked to a wound. We will refer to the patients linked to a wound as the Wound Monitor Cohort. No analysis was conducted on the remaining patients.

In RHIP, 38970 patients have been diagnosed with wounds or a wound-related diagnose. In this article, these patients are referred to as the *RHIP wound cohort*. Patients from the RHIP platform were included if they had at least one ICD-10 diagnose code some time between 2009-2021 that encodes a wound related diagnosis. The ICD-10 codes were defined in the ethical application and include codes for wounds and comorbidities that are known to increase the risk of certain wound types. These comorbidities are venous insufficiency, varicose veins, atherosclerosis and diabetes. All data on patients younger than 18 years old as well as all data from encounters that happened after 2021 was excluded from the RHIP Wound Cohort.

## Method

This article describes the two wound cohorts. This includes investigating various patient and wound variables, their degree of specificity and their prevalence in the data. Results are presented using descriptive statistics. Decisions on relevant results and data interpretation were guided by discussions with clinicians and experts, some of which have been involved in the process of data-generation and/or documentation. The descriptive statistics include demographic and clinical variables as well as the presence and quantity of data subtypes.

The rationale for presenting two cohorts jointly is the complementary perspectives they provide on the wound and patient journey, opening up possibilities for information driven wound care. To deepen the understanding of the potential for information-driven wound care, the differences between the datasets are discussed. The discussion encompasses how and why the datasets were collected as well as their strengths, weaknesses and complementary characteristics.

Because the long-term aim of the research performed on these datasets is to utilize them for AI-modelling, the focus of this cohort profile is not to address biases and distinguish between causal and correlational patterns. Rather, the focus is to describe the datasets in such a way as to deepen the understanding of what types of AI-modelling can be employed on the datasets. The RHIP Wound Care dataset and the Wound Monitor dataset was accessed for research by the authors 2023-10-13 and 2024-01-02 respectively. Both datasets were pseudonymized before they were provided to the authors.

## Ethical statement

The study was approved was for both datasets by the Swedish Ethical Review Authority. For RHIP, no. 2022–00763-01, and for Wound Monitor, no. 2024-07554-01. For RHIP, informed consent was waived and the participants were instead given an opt-out possibility in accordance with the ethical approval. For Wound Monitor, all participants have signed a written consent form for the wound care treatment and the associated processing of their personal data. Mölnlycke has

received only an anonymized dataset from the entity responsible for conducting the wound care treatment. As such, Mölnlycke has not processed any personal data, as defined under the GDPR, in connection with its scientific research.

In the subsequent sections, the results will be presented in accordance with the ethical applications for each of the data sources. This includes making sure no personal data is shared with anyone outside of the stakeholders referenced in the ethical approvals. In addition, after discussions with representatives from Region Halland, it was decided that no results corresponding to fewer than 15 patients would be presented to minimize the risk of identification. As a principle of good practice, the same filtering was applied to the Wound Monitor dataset.

## Results

In this section, descriptive statistics are presented for each of the cohorts. The first subsection focuses on the study populations and includes demographic and clinical variables. The second subsection describes the structure of the datasets including the types and granularity of the data as well as its completeness in relation to the patient and wound journey.

### Study population

Table 1 gives an overview of the demographics of the cohort as well as the prevalence of different wound types. An X indicates that the information is not available.

In the Wound Monitor cohort, the most common wound types are traumatic and surgical wounds. Oncology wounds and infected wounds are the most rare. Patients with at least one diabetic wound has on average 2.18 diabetic wounds (the highest of the wound types). Patients with at least one infected wound has on average 1.17 infected wounds (the lowest of the wound types).

**Table 1**. Overview of cohorts.

|  | Wound Monitor | RHIP |
|---|---|---|
| **Nr of Patients and Wound** | | |
| Nr Patients | 17547 | 38971 |
| Nr Patients with wounds | 17547 | 28696 |
| Nr Wounds | 39890 | X |
| **Sex (n)** | | |
| Male | 7432 (42.4%) | 18512 (47.5%) |
| Female | 10115 (57.6%) | 20459 (52.5%) |
| **Average age at first wound (y)** | | |
| Surgical wound | 63.51 | 63.48 |
| Diabetic wound | 74.36 | 71.47 |
| Leg wound | 77.25 | 76.59 |
| Pressure ulcer | 78.36 | 75.28 |
| Oncology wound | 76.23 | X |
| Traumatic wound | 73.87 | X |
| Infected wound | 61.72 | X |
| Unspecified wound | X | 77.21 |
| Full Cohort | 71.83 | 65.90 |
| **Number of Patients (Number of Wounds)** | | |
| Surgical wounds | 5238 (7419) | 23087 |
| Diabetic wound | 1776 (3878) | 715 |
| Leg wound | 4066 (7497) | 1553 |
| Pressure Ulcer | 5153 (8763) | 2155 |
| Oncology wound | 396 (478) | X |
| Traumatic Wounds | 6265 (10952) | X |
| Infected Wounds | 774 (903) | X |
| Unspecified wound | X | 5502 |

Out of 28696 patients with wounds in the RHIP wound cohort, 23087 have surgical wounds and 5502 have unspecified wounds. A substantial proportion, 10 275 (26%), have no wounds.

The average age at which the first wound is documented is 71.8 and 65.9 in the Wound Monitor Cohort and the RHIP Wound Cohort respectively. The first surgical wound occurs on average at a younger age than Diabetic Wounds. Diabetic Wounds in turn makes the first appearance at a younger age than Leg Wounds and Pressure Ulcers.

The ratio of women to men in the RHIP wound cohort is 1.1:1. This is higher than in the RHIP platform in general, which has a ratio of 1.03:1 [9]. The same ratio in the Wound Monitor dataset is 1.36.

Fig 1 shows the five most common comorbidities in the two cohorts.

In both datasets, essential hypertension is the most common co-morbidity across all wound types except diabetic wounds where type 2 diabetes is more common. The average co-occurrence ratios in RHIP are higher, this is mainly because the ICD-10 data is more complete as described in the section "Database structure and description". Surgical wounds, oncology wounds and infected wounds are less correlated with the most common comorbidities in the Wound Monitor Cohort. In the RHIP Wound Cohort, venous leg ulcers are less correlated with the most common comorbidities than other wound types.

**Demographics.** Fig 2 shows the age at which the first wound appears in the data across the wound types for men and women.

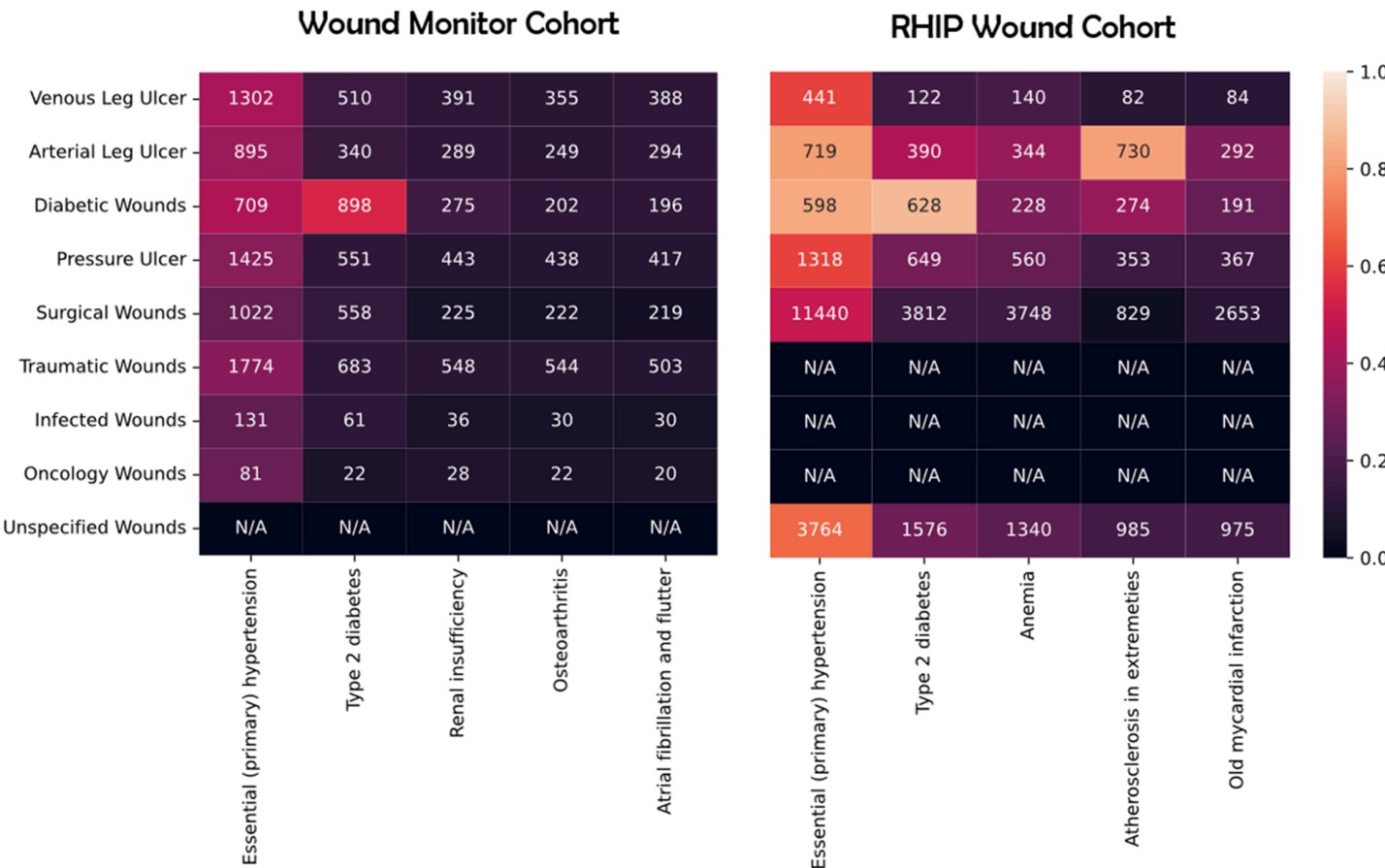

**Fig 1**. **Heatmaps showing the most common comorbidities for the respective databases.** Note: Every number lower than 15 has been changed to 0 due to privacy considerations

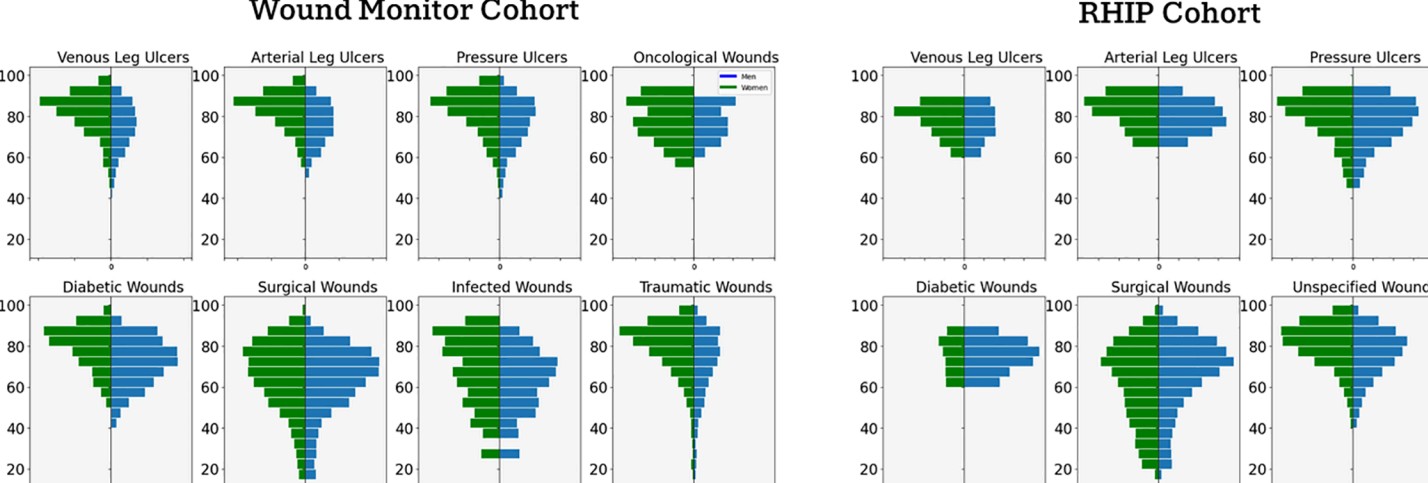

**Fig 2**. **Histograms showing the proportion of patients in different age spans across wound types and sex.** The area per sex and wound type sums to 1.0. Bars corresponding to less than 15 patients have been changed to 0 due to privacy considerations.

Venous leg ulcers (VLUs) are more prevalent among women in both datasets. Gethin et al. (2021) [10] noted that VLUs are more common in women than in men. However the ratio in their study was 1.2:1 while the data from RHIP shows a ratio of 1.4:1 and Wound Monitor 1.7:1. In both datasets, the tilt towards a higher proportion of females in VLUs increases with age up until at least 80-85 year olds.

Arterial leg ulcers (ALUs) are evenly distributed across the sexes in the RHIP dataset, with a slightly lower maximum frequency for men. However, ALUs are more common in women in the wound monitor dataset, the ratio being 1.4:1. No comparison of male to female ratio for arterial leg ulcers was found in the literature.

Pressure ulcers are also evenly distributed across the sexes in the RHIP dataset. In the Wound Monitor dataset, the ratio of women to men is 1.3:1. In a risk-profile of Pressure Ulcers, Gaspar et al. [11] analyzed a population of 2996 patients where the ratio of women to men was 1.4:1.

Diabetic wounds are more common among men in both datasets. Engberg et al. (2019) [12] studied a cohort of 780 patients with diabetic foot ulcers where the ratio of men to women was 2.2:1. Shah et al. 2021 [13] did a demographic profile of a cohort of 75 patients with diabetic foot ulcers where the ratio of men to women was 3.4:1. The RHIP Wound Cohort has a ratio of men to women of 2.5:1. In the RHIP Wound Cohort, the ratio of men to women in the number of patients with diabetes is 1.4:1. The disproportionate amount of men with diabetes is less significant than in Diabetic Foot Ulcers. This may indicate that men are both more likely to develop diabetes and to develop Diabetic wounds once they have developed diabetes. The Wound Monitor Cohort has a ratio of men to women of 1.4:1.

Oncological wounds are more common in women in the Wound monitor dataset, the ratio being 2.1:1. Surigcal wounds are slightly more common among men in both datasets, with the ratio being 1.1:1 in the Wound monitor dataset. Infected wounds and traumatic wounds are both more common among women in the wound monitor dataset, with the ratios being 1.2:1 and 1.7:1 respectively.

Diabetic wounds, leg ulcers, pressure ulcers and traumatic wounds most commonly occur somewhere between age 70 and 90. This maximum frequency varies between men and women. Among these wound types, diabetic foot ulcers peak the earliest, around age 70-85. The other types peak over the age of 80. SBU (2014 s.41) [3] noted that diabetic foot ulcer patients are younger than the rest of the chronic wound-population. On average, men have their first wound

documented at age 68.0 and 67.2 in Wound Monitor and RHIP respectively. For women, the corresponding averages are 74.7 and 64.6.

## Database structure and description

The following subsection outlines the structure and content of the databases.

**Content, quality, quantity and granularity.** In both databases, each data subtype (e.g. diagnosis, procedures, medication) is stored in a relational table where patient ID is matched to a date and a coded event. As an example, for the diagnosis table, each row contains a patient ID, a date and the associated ICD-10 code. The combination of all the relational tables enables patient journey analysis.

In Table 2, the subtypes of patient journey data is presented for each database respectively. The coverage, defined as the proportion of patients with at least one event described in this data subtype, is also presented. Lastly, it shows the number of events per patient across the data subtypes.

The coverage is close to 100% in the RHIP database. This is a consequence of the RHIP data being populated directly from the healthcare organization's documentation systems. The patient level data can therefore be assumed to contain all data that is digitally documented through the official systems in these organizations. Diagnoses, Medications and Procedures does not come directly from the health care system in the Wound Monitor Database. Instead, it is acquired through a combination of registries, referrals and patient dialogue. Therefore, this data is not complete as can be observed in Table 2. In both datasets, diagnoses are described with ICD-10 codes, Medications are described with ATC-codes. For the RHIP database, procedures are described with KKÅ codes. The procedures in Wound Monitor are not described with a coding system but just a description of the procedure in plain text. However, these descriptions have a common pattern, enabling structured analysis.

The vital signs data in both databases ranges from bedside measurements such as glucose concentration and physical measurements such as pulse or height. The vital signs data in the Wound Monitor Cohort has a predefined small set of measurements that are performed on most patients. There is a large variation of how many of these variables are documented. Both the vital signs and the laboratory data in the RHIP wound cohort are sparse. Additionally they are unstructured from a data analysis perspective. Rather than having a predefined set of variables, a measurement is performed due to a doctor considering it to be relevant to a diagnosis or a suspected diagnosis. Given that much of the diagnosis data is excluded from the original data source based on the ethical application this is difficult to analyze in relationship to diagnosis data.

The wound monitor dataset contains quality of life assessments and nutritional information. The quality of life assessments have been conducted using the EQ-5D-3L tool developed by the Euroqol group [14]. The nutrition data contains

**Table 2**. Data content comparison for patient data.

| | Wound Monitor | | RHIP | |
|---|---|---|---|---|
| | **Coverage** | **Events per patient** | **Coverage** | **Events per patient** |
| Diagnosis | 77% | 10.18 | 100% | 116.67 |
| Medication | 81% | 16.38 | 98% | 537.06 |
| Procedures | 66% | 6.85 | 97% | 41.69 |
| Vital signs | 85% | - | 99% | - |
| Laboratory values | N/A | N/A | 99% | - |
| Quality of life | 43% | 1.89 | N/A | N/A |
| Nutrition | 68% | 2.01 | N/A | N/A |

Data subtypes in relation to the number of patients for which information is available (*Coverage*) as well as the frequency (*Events per patient*). N/A indicates that the data type is doesn't exist in the database and therefore the value is not available. '-' indicates that a value hard to define due to structural reasons.

two categorical variables. One variable describes if a patient is given help with feeding themselves and if so how that what type of help. The second variable describes the degree of nutritional sufficiency.

The RHIP dataset is more complete at patient-level than the Wound Monitor dataset. The reverse is true for data at wound-level. Table 3 gives an overview of the type of variables that are captured in the documentation of wounds. A detailed comparison is given in the next subsection where the classification of the wounds is described in depth.

Patients have almost 10 times as many wound-related visits in the wound monitor dataset. This observed imbalance is most likely due to that RHIP does not capture home-care visits where the majority of wound care occurs. Naturally, there is also a much longer time between these visits, with an average of almost 100 days. In Wound monitor the interval is approximately a week on average. The most common visit-pattern in this dataset is weekly visits.

The tabular part of RHIP only contains classification and location of the wound. The location of the Diabetic wounds and leg wounds is specified by the ICD-10 code. Surgical Wounds and Pressure Ulcers have an optional specification for location in the ICD-10 codes but this is not always used. The resolution of the location-description is substantially higher in the Wound monitor dataset.

The RHIP Wound Cohort does not contain the identification of a wound, i.e. it is not indicated if two different health-care visits concerns the same or different wounds. Moreover, there are no explicit outcome variables such as a healed wound that could be used directly for predictive modelling. Further, there are no variables describing the characteristics of the wound such as periwound skin, edge, tissue type, surface area. In the RHIP database there are clinical notes linked to most visits. Right now, the authors of this paper do not have acces to these notes. However, there is ongoing work to extract relevant facts from the notes utilizing traditonal NLP methods as well as Large Language Models. It is possible that some of the variables described as missing above could be extracted. Discussion with clinicians have indicated that variables such as location, wound healing, and tissue are often documented in free text, at least when considered relevant. This means these variables that its likely that these variables can be extracted for some visits. Wound identification might be extracted by references in the clinical notes such as "there is less granulation than last week". Extraction of variables from free-text is inherently uncertain and therefor the authors intend to use methods for uncertainty quantification during this process.

The Wound Monitor Cohort contains all of the variables mentioned in the previous paragraph and has on average 15.93 visits documented per wound. This means that a specific wound can be identified and tracked over the course of progression in relation to specific target variables. The Wound Monitor database also contains a Braden Scale risk assessment [15] for 1335 of the patients with pressure ulcers. This is not linked with a wound identification variable but only a patient idenfitication variable. Since quite few patients have multiple pressure ulcers at the same time this for the

**Table 3**. Data content comparison for wound data.

| | Wound Monitor | RHIP |
|---|---|---|
| Average number of wound-related visits per patient | 35.9 | 3.5 |
| Average number of days between wound-related visits | 8.6 | 116.4 |
| Healing Wound | Yes | No |
| Wound ID | Yes | No |
| Surface Area | Yes | No |
| Location | Yes | Occasionally |
| Periwound Skin | Yes | No |
| Tissue type values | Yes | No |
| Treatment | Yes | No |
| Treatment Plan | Yes | No |
| Images of wound | Yes | No |

The type of wound data included in the two databases. The average number of wound related visits per patients is calculated using only the number of patients with wounds, and not the full cohort, as the denominator.

most part not problematic. Additionally, it contains information on treatment plans for each wound as well as which products have been used each visit.

Fig 3 shows an example of what typically would be documented for the same patient in the two databases. The example illustrates commonalities and differences in content, quantity, quality and granularity from a patient journey perspective as well as a wound journey perspective based on the results presented in this section.

**Wound classification.** This section describes how wounds are classified in the respective databases. Because different systems were used in the respective cohorts for the classification of wounds, no direct comparison between the different subcategories is made, except for pressure ulcers were the same system is used occasionally.

**Wound monitor.** The wound monitor dataset classifies the wounds into seven main categories. Diabetic Wounds, Pressure Ulcers, Oncology Wounds, Traumatic Wounds, Infected Wounds, Leg Wounds and Surgical Wounds. Each of these categories, except oncology wounds, are further classified into one or two levels of subcategories as shown in Fig 4.

The *diabetic wounds* are classified with the University of Texas (UT) diabetic wound classification. This is a classification system assessing depth, presence/lack of presence of infection and presence of signs of lower extremity ischemia. The UT system has two axes, the first axis corresponds to 4 levels of depth (Grade 0-3). The second axis corresponds to all four possible combinations of presence of infection and clinical signs of ischemia [16]. This can be seen as two binary variables for analytic purposes. The *pressure ulcers* are classified according to the (National Pressure Injury Advisory Panel) NPIAP system. This means that the wounds are classified into 4 possible stages of ulcer depth as well as a fifth class which corresponds to non-classifiable depth [17]. A small portion of the pressure ulcers are not classified according to this system but are instead classified as either suspected deep tissue (n = 125) or incontinence/moisture damage (n = 250). The ulcers that are in the classified as incontinence/damage are further classified into Mild, Moderate and Severe Incontinence-associated dermatitis (IAD), fungal infection. The *traumatic wounds* are classified based on the cause of the wound. The classes are laceration, disrupted skin integrity, abrasion, bruise, abscess, avulsion, bite wound,

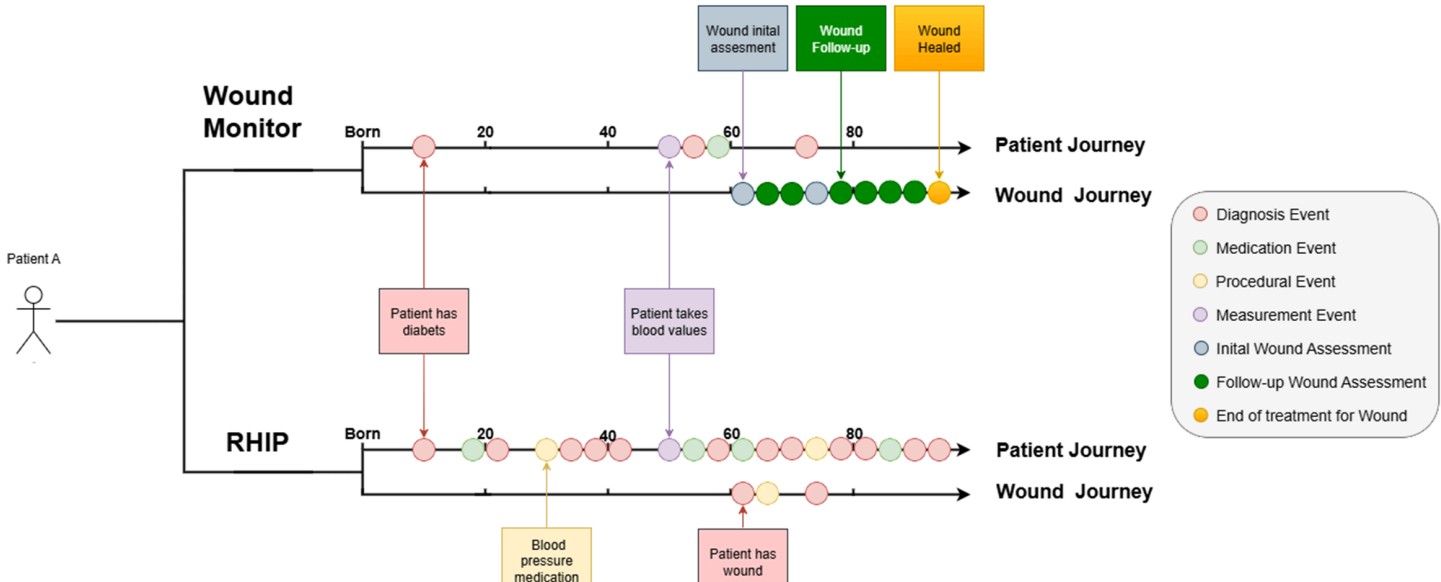

**Fig 3**. A fictional example of a what is typically being captured with regards to Patient and Wound Journey in the respective databases.

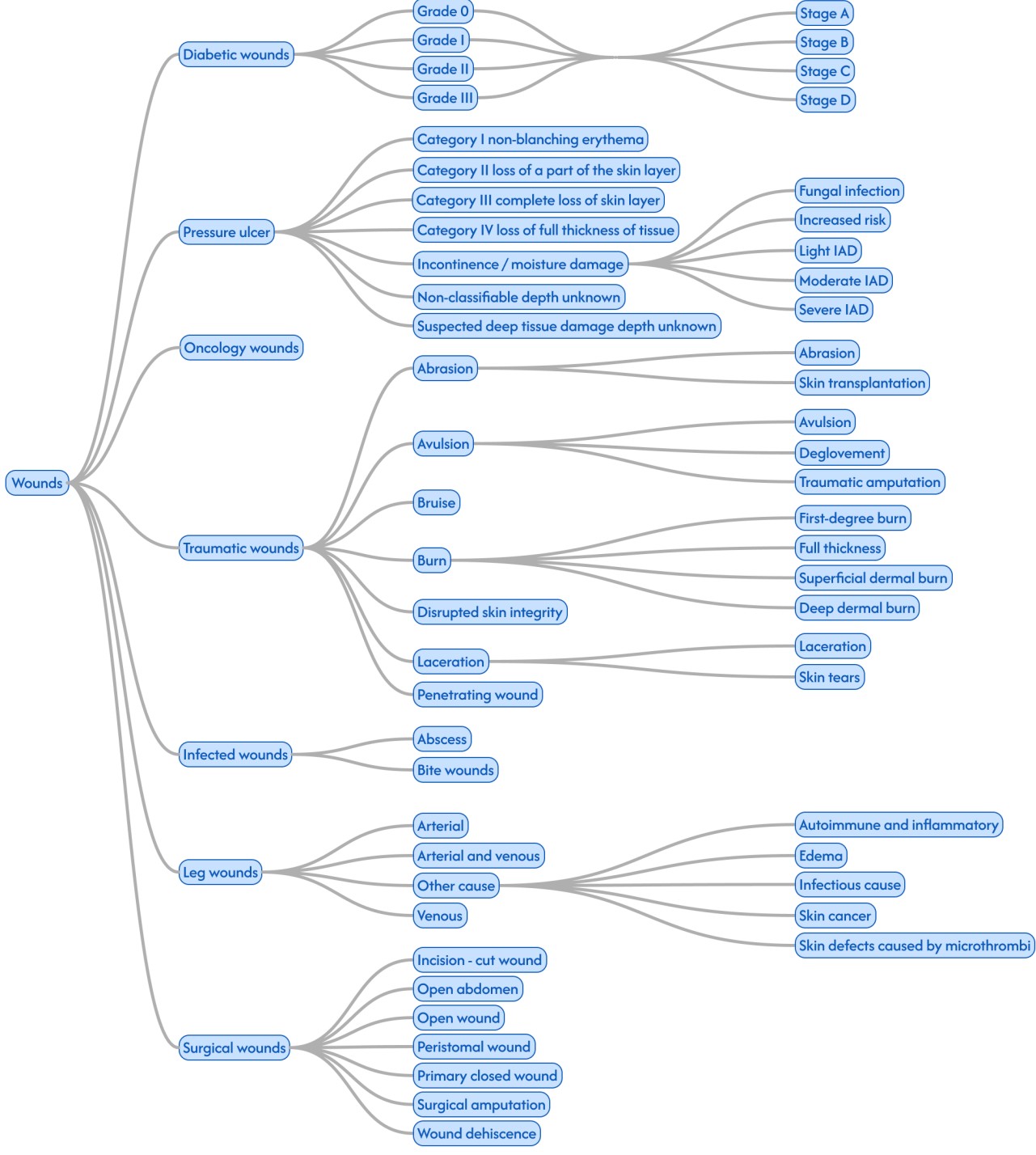

**Fig 4. Wound classification tree for the Wound Monitor database.** The categories traumatic wounds, infected and surgical wounds were originally part of the same category while burns were a category of their own. Discussions with clinicians lead to the structure seen in the figure.

petetrating wound and burn. The burns are further classified into four categories of thickness according to the classification system of the american burn association [18]. The *infected wounds* are classified into abcess and bite wounds. The *leg wounds* are classified based on the cause. The causes are venous, arterial, venous and arterial, edema, infection, autoimmune and inflammatory and microthrombi. For some leg wounds, the cause is unknown (n = 402). The *surgical wounds* are classified based on either the type of surgery performed (incision, amputation, open abdomen) or the state of the wound after the surgery (peristomal wound, dehiscence, open wound, primary closed wound). Fistula is also included since, though it's not surgical per se, it has similar characteristics.

**RHIP.** The wounds in the RHIP dataset are classified with ICD-10 codes. As can be observed in Fig 5, the ICD-10 classification of wounds is not granular. There are two codes for venous leg ulcers, one that indicates leg ulcer (I830), and one that indicates leg ulcer with inflamation (I832). Arterial leg ulcer has one diagnose-code (I172C). There are three diagnose codes for diabetic wounds: Diabetes type 1 with foot ulcer (E106D), Diabetes type 2 with foot ulcer (E116D) and Diabetic osteopathies in foot (M908H). For pressure ulcers there are 15 diagnose codes. Four of them are called grade 1-4 but they align with are the four categories from the NPIAP guidelines described above. The remaining eleven codes signify the location of the wound and these are not utilized regularly in the RHIP Wound Cohort. Because of the richness of information on the wound location in the Wound Monitor Cohort it was decided to not include location-specification in

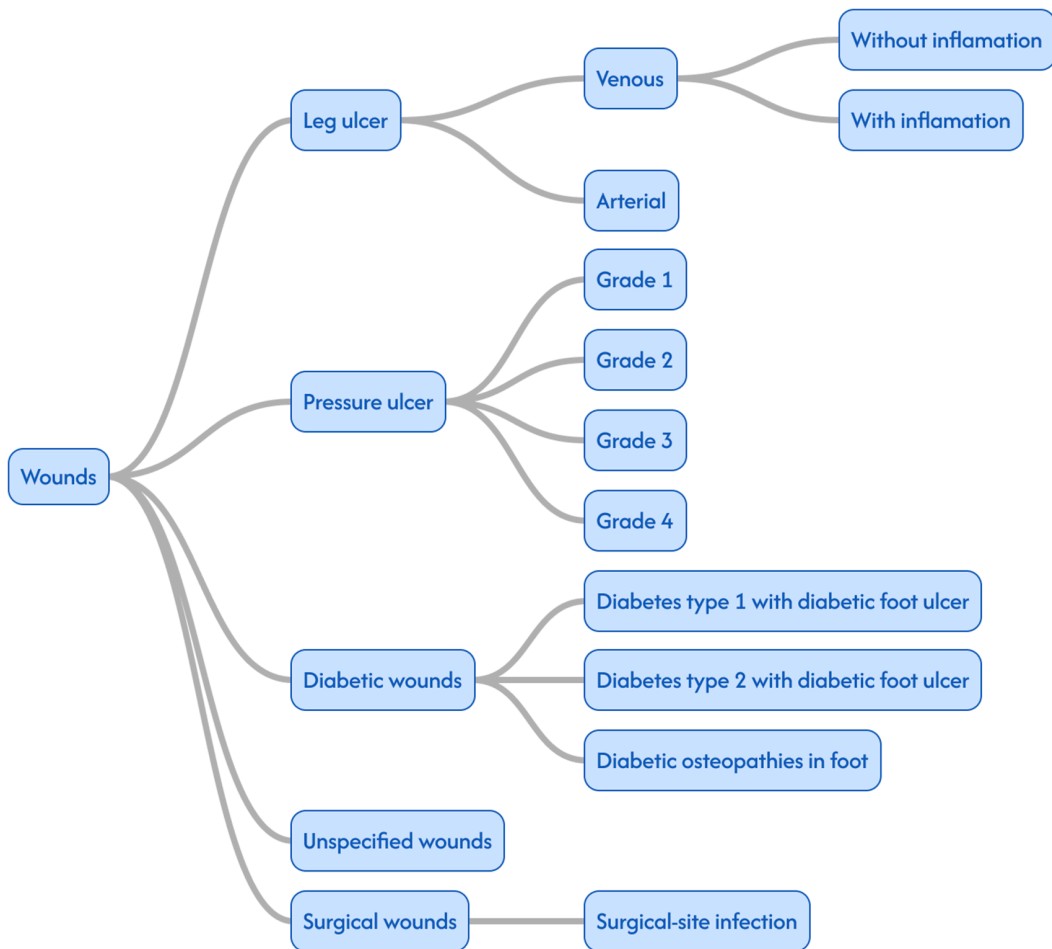

**Fig 5. Wound classification tree for the RHIP Wound Cohort.**

the classification trees. The surgical wounds have many diagnose codes but they are not defined from a wound perspective. Instead they describe post-surgical complications more generally.

## Discussion

The databases have been created for different purposes which is reflected in their informational content. Therefore, their potential for being utilized in modelling of patient- and wound journey differs. This section is divided into three subsections, the first describes the various strategies for handling data quality throughout the various analysis. This was left out of the method section because of the need to reference different figures and tables of the result section. The second discusses the strengths and weaknesses of the datasets while the last discusses the potential of utilizing these datasets in AI-modelling in the service of information-driven wound care.

### Data cleaning, harmonization and handling of incomplete data

The databases are presented in their original structure apart from a small adjustment in Wound Monitor described in the caption to Fig 4. In order to compare wound types between the cohorts in Table 1, the ICD-10 codes in RHIP for arterial leg ulcers and venous leg ulcers were aggregated into one category, *leg wounds.* This was done to include all leg wounds from Wound Monitor since a some wounds in this cohort are classified with another cause than arterial or venous insufficiency. In Figs 2 and 3, the opposite harmonization was done and leg wounds in the Wound Monitor cohort were split up into arterial leg ulcers and venous leg ulcers since the differences in the profile of these patients was of interest. The leg wounds in Wound Monitor that were neither arterial or venous were left outside of Figs 2 and 3. Apart from this, no harmonization of wound types was needed since the the cohorts either shared categories (diabetic wounds, pressure ulcers and surgical wounds), or did not overlap at all (oncology wounds, traumatic wounds, infected wounds and unspecified wounds). Real world data is always going to be more or less incomplete with regards to essential information. The datastructures of the databases presented in this article are for the most part event-based and as a consequence such incompleteness is hard to identify. For example it is hard to tell if a patient with venous insufficiency if no such information has been documented. However there are a few exceptions, as demonstrated in Fig 1, there are patients with diabetic wounds with no diabetes documented. Here it might be reasonable to assume that the documentation of the wound is correct and the health care professions was informed that the patient had diabetes though that was never documented. For this article, it was decided to not make any such assumptions but just presenting the data as it is. However, the authors aim to give an sense of the degree of incompleteness of the databases by presenting the number of events documented per patient for various data subtypes as well as the number of patients with at least one event documented, see Table 2. *Nan* values encountered in measurements, diagnosis, procedures et.c. were removed from the analysis presented in Tables 2 and 3. Apart from this no cleaning of data was needed for the results presented in the article. There were no missing values for the high level demographics presented in Table 1.

### Strengths and weaknesses

**RHIP wound cohort - Strengths.** The main strength of the RHIP Wound Cohort lies in its potential for a patient journey analysis. There are a multitude of reasons for why it has this potential.

The fact that the data comes directly from the documentation system in Region Halland, combined with the level of integration between different parts of the healthcare systems, means that the RHIP Wound Cohort enables a holistic insight into the patient journey. This is reflected in the coverage and number of events per patient shown in Table 2. Additionally, the level of integration provides a detailed description of a patients journey across care organizations, e.g. primary care centers (both public and private), specialist care at hospitals, and emergency departments. Given that it is established that a main-driver of cost in wound care is the lack of the right care at the right time, this might prove to be of great value in future work.

 

**RHIP wound cohort- Weaknesses.** Wound care is not prioritized in the healthcare system [3]. This may be one of the reason why documentation of wounds is not done optimally in relation to a wound journey analysis.

As shown in Table 3, the frequency of wound-related visits is low and many of the key variables for wound analysis are missing. Two of the most important variables that are missing are *Wound identification* and *Reason for ending treatment*. Without a *Wound identification* variable it is not possible to tell if the same wound is being treated at subsequent visits. *Reason for ending treatment* is important in order to distinguish between treatments that work and treatments that do not work. Additionally, variables tracking the progress of the wound are also necessary for an in depth wound journey analysis. This could for example be wound size or skin characteristics. These challenges might potentially be harnessed by the utilization of free-text data in future work.

As illustrated in Fig 5, the classification of wounds in the RHIP Wound Cohort is not granular. In contrast to the Wound Monitor Cohort, few of the international classification tools are being deployed. The exception is Pressure Ulcers where the NPIAP guidelines are used in some cases. In addition, as shown in Table 1, the second most common wound is *undefined wound*. The lack of granularity and specificity in the wound classification poses a challenge in machine learning modelling of different trajectories in wound journeys.

A weakness from a patient journey perspective in the analysis of the RHIP Wound Cohort is that most medications are supplied by pharmacies and not at hospitals. In the latter case, the documentation of the medicine does not include a visit identification that can be utilized to link certain medication to a specific diagnose code.

Lastly, a challenge in using the RHIP Wound Cohort for wound journey analysis is that wounds are mostly treated in home-care and elderly care. This means a lot of key information is outside the scope of the database.

**Wound monitor cohort - Strengths.** The Wound Monitor Cohort has many strengths from a wound journey perspective. One strength is its detailed description of wounds, as shown in Fig 4 and Table 3. Another strength is the tracking of wound progress. As shown in Table 3 the cohort contains multiple important variables. The large collection of wound images also opens the potential for image analysis and multi-modal machine learning approaches.

The Wound Monitor dataset also contains important patient variables that are not present in the RHIP Wound Cohort such as Quality of Life estimations and nutritional intake as shown in Table 2.

**Wound monitor cohort - Weaknesses.** The main weakness of the Wound Monitor is that it is not complete in description of patients characteristics as shown in Table 2. A clear indicator of the lack of completeness is shown in Fig 1 where the correlation coefficients for the RHIP Wound Cohort is substantially higher on average between wounds and comorbidities. We can assume that most of the difference in correlations of wounds and co-comorbidities between the cohorts is not driven by an actual difference in the state of the patients but rather by a difference in completeness of information.

## Future work

We intend to train and explore both shallow and deep machine learning models to characterize and predict events in the patient journey based on the RHIP Wound Cohort. Optimally, the predictive power and model inner-workings will illuminate how different healthcare decisions lead to different clinical and cost outcomes. We intend to utilize the Wound Monitor Cohort for characterization of the wound journey as well as creating models for predicting wound healing and wound state.

In addition to standard machine learning approaches, we intend to utilize conformal prediction and eXplainable AI frameworks in order to make the models trustworthy both in terms of explainability and statistical guarantees.

**Cross-cohort analysis and information-driven wound care.** So far, it has been established that the RHIP Wound Cohort can be used for modelling the patient journey and the Wound Monitor Cohort can be used for wound journey modelling. This subsection contains a discussion about how the datasets might be used in a complementary fashion in the service of information-driven wound care.

As shown in Fig 3, the Wound Monitor cohort provides a detailed description of the trends in a wound while in the RHIP Wound Cohort there is high granularity in describing trends prior to a wound as well as patient journey trends parallel to the wound journey. Provided that the insights from one dataset translates to the other dataset it is therefore possible to cover a large space in the landscape of opportunities in information-driven wound care. As an example, models built based on the Wound Monitor Cohort could be utilized to help the process of wound documentation in the future of the RHIP database. Such cross-cohort insights align with the framework of information-driven wound care.

Table 1 gives insight into similarities and differences between the cohorts. The main difference in the demographics between the cohorts is that the wound monitor dataset contains older patients and more women, as seen in Table 1. The main reason for the patients being older is most likely because patients in home-care are older than people with wounds in general. The disparity between men and women might partly be explained by the fact that women live longer than men. However, it is also the case in these datasets, as can be seen in Fig 2, that women tend to get their wounds later than men across all wound types. There is likely a large overlap in patient and wound characteristics for the patients in the two cohorts that have the same wound types. The relative prevalence of the chronic wound types and the relative difference in age at first wound is similar in the two dataset which might indicate overlapping population-level characteristics between these patients and wounds. Fig 1 shows that hypertension and diabetes is the two most common co-morbities in both cohorts. A notable difference is the co-occurrence of venous leg ulcers with different comorbidities being substantially lower in the RHIP Wound Cohort while no such trend can be observed in the Wound Monitor Cohort. It is difficult to compare the comorbidities in the two cohorts in detail. This is due to the RHIP Wound Cohort being limited by the diagnose codes listed in the ethical application and the Wound Monitor Cohort diagnose data being far from complete. Therefor, the lack of overlap comorbidities does not necessarily indicate a difference between the cohorts. The generalizability of the findings to other populations can be argued. Given the impact of age, health conditions and lifestyle on wound healing, the cohorts are not representative of all populations. Further, the Wound Monitor database likely has a socio-economic bias since some people can not afford to prioritize private wound care. Additionally, some insights into the documentation process might not be immediately applicable in contexts with less digital infrastructure. However insights into improvements of the wound care workflow are still relevant across all populations.

## Conclusion

This article article has described two wound cohorts and their respective demographic and clinical variables. The RHIP Wound Cohort has more than twice as many patients though only about three quarters of them have wounds. The patients are on average older and more predominantly female in the Wound Monitor Cohort. The distribution of patients with chronic wounds is similar while the Wound Monitor cohort have some additional categories and no unspecified wounds. The purpose of the healthcare organization and its data-collection process influences the structure, content and quality of the data. The two databases described in this article have been generated in different contexts. As a result, the type of information they contain as well as the potential they have for being utilized in advanced analytics and machine learning differs. Broadly, the RHIP Wound Cohort is more fit for a patient-journey perspective while the wound monitor Cohort is more fit for a wound journey perspective. Together, they create a solid foundation for a comprehensive approach towards information-driven wound care.

## Author contributions

**Conceptualization:** Oskar Gustafsson, Jens Lundström, Mattias Ohlsson, Hanna Stenhamre, Daniel Tsang, Ernst Ahlberg.

**Data curation:** Oskar Gustafsson.

**Formal analysis:** Oskar Gustafsson, Jens Lundström, Mattias Ohlsson, Daniel Tsang, John Pavia, Ernst Ahlberg.

**Funding acquisition:** Ernst Ahlberg.

**Investigation:** Oskar Gustafsson, Ernst Ahlberg.

**Methodology:** Oskar Gustafsson, Jens Lundström, Mattias Ohlsson, Hanna Stenhamre, Daniel Tsang, Ernst Ahlberg.

**Project administration:** Oskar Gustafsson, Jens Lundström, Mattias Ohlsson, Ernst Ahlberg.

**Resources:** Oskar Gustafsson, Jens Lundström, Mattias Ohlsson, Ernst Ahlberg.

**Software:** Oskar Gustafsson.

**Supervision:** Oskar Gustafsson, Jens Lundström, Mattias Ohlsson, Hanna Stenhamre, Ernst Ahlberg.

**Validation:** Oskar Gustafsson.

**Visualization:** Oskar Gustafsson.

**Writing – original draft:** Oskar Gustafsson.

**Writing – review & editing:** Oskar Gustafsson, Jens Lundström, Mattias Ohlsson, Hanna Stenhamre, Daniel Tsang, John Pavia, Ernst Ahlberg.

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
