## [Decision Letter · Decision Letter 0]

26 Aug 2025

PONE-D-25-23383Cohort Profile: The Dutch Wound Monitor Cohort and the Swedish Region Halland Integrated Platform (RHIP) Wound CohortPLOS ONE

Dear Dr. Gustafsson,

Thank you for submitting your manuscript to PLOS ONE. After careful consideration, we feel that it has merit but does not fully meet PLOS ONE’s publication criteria as it currently stands. Therefore, we invite you to submit a revised version of the manuscript that addresses the points raised during the review process.

We look forward to receiving your revised manuscript.

Kind regards,

Yih-Kuen Jan, PhD

Academic Editor

PLOS ONE

“This work was conducted as part of the CAISR Health Research Profile at Halmstad University with funding made possible by the Knowledge Foundation (grant number 20200208 01H).”

“This work was conducted as part of the CAISR Health Research Profile at Halmstad University with funding made possible by the Knowledge Foundation (grant number 20200208 01H).”

“This work was conducted as part of the CAISR Health Research Profile at Halmstad University with funding made possible by the Knowledge Foundation (grant number 20200208 01H).”

Reviewers' comments:

Reviewer's Responses to Questions

**Comments to the Author**

1. Is the manuscript technically sound, and do the data support the conclusions?

Reviewer #1: Yes

Reviewer #2: Yes

2. Has the statistical analysis been performed appropriately and rigorously?

Reviewer #1: Yes

Reviewer #2: Yes

3. Have the authors made all data underlying the findings in their manuscript fully available?

Reviewer #1: Yes

Reviewer #2: Yes

4. Is the manuscript presented in an intelligible fashion and written in standard English?

Reviewer #1: Yes

Reviewer #2: Yes

5. Review Comments to the Author

Reviewer #1: Dear Authors, thank you very much fir the article. The manuscript is clearly structured, well-written, and provides useful demographic and clinical insights into two distinct but complementary data sources.

The detailed presentation of database characteristics, strengths, and weaknesses is helpful for potential secondary users and for researchers considering cross-cohort analyses. However, there are aspects that warrant clarification or improvement to strengthen the manuscript’s rigor, contextualization, and utility.

Please consider this Major Points for Consideration

The article is positioned as a “cohort profile,” yet also discusses potential AI modelling applications. While this is valuable, the primary purpose (descriptive vs. preparatory for modelling) should be explicitly clarified in the Introduction and Conclusion.

Consider please, stating more clearly how these datasets are expected to contribute to improving wound care workflows or informing policy.

Contextualization and Literature Integration should be improived since i fekt the need to better situate these cohorts within the broader landscape of wound databases internationally. Are there similar registries (e.g., U.S., U.K., Australia) and how do these compare in scope or structure?

Please clarify or explain the rationale for comparing Dutch and Swedish data (beyond availability) should be explained, particularly regarding differences in healthcare delivery and wound care practices.

Methodological Detail and Transparency

We fekt the need to have a better description or more detail on data cleaning, quality control, and missing data handling is warranted. For example, how were errors or inconsistencies in wound classification addressed? Please, clarify whether any harmonization of variable definitions was attempted between cohorts for comparative purposes.

Regarding Ethics and Data Access, and we noticed that ethical approvals are listed, but we consider that the manuscript should explicitly address data privacy measures, especially given the use of real-world clinical data and potential re-identification risks in smaller subgroups. - just a reflection

Last suggestuoins are associated with Limitations and Biases, because while some dataset limitations are acknowledged, additional discussion of selection bias (e.g., home-care patients in the Dutch cohort vs. broader care settings in RHIP) would contextualize findings.

With these revisions to clarify objectives, strengthen contextualization, and expand discussion of methodological considerations, the paper would provide a valuable resource for researchers and clinicians in wound care and digital health.

Reviewer #2: This is a well-structured and timely manuscript that provides an in-depth cohort profile of two valuable wound care datasets: the Dutch Wound Monitor and the Swedish RHIP Wound Cohort. The manuscript is technically sound, and the conclusions are appropriately supported by the data presented.

The strengths of the study include:

Clear and meaningful comparison between patient-level and wound-level data perspectives.

Integration of demographic, clinical, and data infrastructure details.

Transparent methodology and ethical compliance.

A forward-looking discussion about AI potential in wound care.

The paper makes a strong case for how real-world data can enable information-driven care in chronic wound management.

Suggestions for minor revision:

1. Enhance the clarity of Figure 3 – the contrast between RHIP and Wound Monitor could be more intuitive to general readers.

2. Include a concise summary table comparing the wound classification systems used in both cohorts (e.g., ICD-10 vs. NPIAP/UT).

3. Briefly address the generalizability of findings outside Sweden and the Netherlands, especially in regions with limited digital infrastructure.

4. Clarify how future NLP/LLM efforts might resolve missing wound ID and outcome variables in RHIP.

These revisions are relatively minor and aimed at improving clarity and impact. Overall, I support the publication of this work and believe it will be of value to the wound care, data science, and digital health communities.

6. PLOS authors have the option to publish the peer review history of their article (what does this mean?). If published, this will include your full peer review and any attached files.

Reviewer #1: **Yes:** Paulo Jorge Pereira Alves

Reviewer #2: **Yes:** Abdulrahman Almalki

---

## [Author Response · Author response to Decision Letter 1]

4 Nov 2025

Response to Reviewers

We thank the Academic Editor and reviewers for their thoughtful and constructive feedback. Below, we address each comment in detail and describe the corresponding changes made to the manuscript.

Academic Editor Comments

Comment 1: Please ensure that your manuscript meets PLOS ONE’s style requirements, including those for file naming.

Author Response: Thank you for this comment. We have reviewed the style requirements and ensured that our manuscript and files comply with them.

Comment 2: Thank you for stating the following financial disclosure: “This work was conducted as part of the CAISR Health Research Profile at Halmstad University with funding made possible by the Knowledge Foundation (grant number 20200208 01H).” Please state what role the funders took in the study. If the funders had no role, please state: “The funders had no role in study design, data collection and analysis, decision to publish, or preparation of the manuscript.” If this statement is not correct, please amend it as needed. Include the amended Role of Funder statement in your cover letter; we will change the online submission form on your behalf.

Author Response: Thank you for your comment. We have included the amended Role of Funder statement in the cover letter. We appreciate your assistance in updating the online submission form.

Comment 3: Thank you for stating the following in the Acknowledgments Section of your manuscript: “This work was conducted as part of the CAISR Health Research Profile at Halmstad University with funding made possible by the Knowledge Foundation (grant number 20200208 01H).” We note that you have provided funding information that is currently declared in your Funding Statement. However, funding information should not appear in the Acknowledgments section or other areas of your manuscript. Please remove any funding-related text from the manuscript and let us know how you would like to update your Funding Statement. Currently, your Funding Statement reads: “This work was conducted as part of the CAISR Health Research Profile at Halmstad University with funding made possible by the Knowledge Foundation (grant number 20200208 01H).” Please include your amended statements within your cover letter; we will change the online submission form on your behalf.

Author Response: Thank you for your feedback. We have removed the funding-related text from the manuscript and updated the cover letter with an amended Funding Statement.

Comment 4: Please review your reference list to ensure that it is complete and correct. If you have cited papers that have been retracted, please include the rationale for doing so in the manuscript text, or remove these references and replace them with relevant current references. Any changes to the reference list should be mentioned in the rebuttal letter. If you need to cite a retracted article, indicate its retracted status in the References list and include a citation and full reference for the retraction notice.

Author Response: We have added three new references (numbers 6, 7, and 8) in the revised manuscript. These are cited in a new paragraph in the Introduction section, where we situate the cohorts within the broader landscape of cohort studies, as per Reviewer 1’s comment.

Reviewer 1 Comments

Comment 1: The article is positioned as a “cohort profile,” yet also discusses potential AI modelling applications. While this is valuable, the primary purpose (descriptive vs. preparatory for modelling) should be explicitly clarified in the Introduction and Conclusion. Please state more clearly how these datasets are expected to contribute to improving wound care workflows or informing policy.

Author Response: Thank you for pointing this out. We have clarified that the primary aim is to characterize the two wound cohorts, and that the discussion of AI modeling is a forward-looking reflection. We revised the abstract, introduction, and conclusion to make this clearer. These changes appear in the last seven lines of the abstract, the last three lines of the third paragraph in the introduction, and the first six lines of the conclusion.

Comment 2: Regarding Ethics and Data Access: while ethical approvals are listed, the manuscript should explicitly address data privacy measures, especially given the use of real-world clinical data and potential re-identification risks in smaller subgroups.

Author Response: We appreciate this reflection. In the revised manuscript, we added a paragraph detailing the measures taken to ensure data privacy. This paragraph is now the final paragraph of the Ethical Statement section. It includes content moved from the Inclusion and Exclusion Criteria section and new material added in response to your comment.

Comment 3: Please clarify the rationale for comparing Dutch and Swedish data (beyond availability), particularly regarding differences in healthcare delivery and wound care practices.

Author Response: Thank you for this comment. While the authors are from Sweden and the Netherlands, the rationale for presenting these two cohorts together is that they offer complementary perspectives on the patient and wound care journey. We have clarified this in the revised manuscript, specifically in the first sentence of the second paragraph in the Methods section.

Comment 4: We felt the need for a better description or more detail on data cleaning, quality control, and missing data handling. For example, how were errors or inconsistencies in wound classification addressed? Please clarify whether any harmonization of variable definitions was attempted between cohorts for comparative purposes.

Author Response: This is a good point. We have now clarified our approach to quality control, data cleaning, missing data handling, and harmonization. These details are included in the revised manuscript in the Discussion section under the first new subsection. We chose not to place this content in the Methods section due to PLOS ONE’s formatting guidelines, which restrict referencing figures and tables until they are presented. Additionally, some of the rationale for our decisions is more appropriate for discussion rather than methodological description.

Comment 5: Contextualization and literature integration should be improved. We felt the need to better situate these cohorts within the broader landscape of wound databases internationally. Are there similar registries (e.g., U.S., U.K., Australia), and how do these compare in scope or structure?

Author Response: Thank you for this comment. We agree that it is important to describe what is known about other registries. In the revised manuscript, we have added a paragraph in the Introduction section that discusses similar cohorts internationally. This paragraph includes three new references added to the reference list.

Comment 6: Limitations and biases: While some dataset limitations are acknowledged, additional discussion of selection bias (e.g., home care patients in the Dutch cohort vs. broader care settings in RHIP) would help contextualize findings.

Author Response: This is a fair point and should have been addressed in the original manuscript. Another reviewer also requested more reflection on the generalizability of potential findings. We have added a few sentences in the Discussion section to address both generalizability and the selection bias issue. These additions can be found in the last seven lines under the headline “Information driven care and cross-cohort analysis.”

Reviewer 2 Comments

Comment 1: Enhance the clarity of Figure 3 – the contrast between RHIP and Wound Monitor could be more intuitive to general readers.

Author Response: Thank you for this comment. We agree that the contrast between the databases was not clearly demonstrated. We have edited Figure 3 to make the differences and similarities between RHIP and Wound Monitor more intuitive for general readers. The figure remains labeled as Figure 3 in the revised manuscript.

Comment 2: Include a concise summary table comparing the wound classification systems used in both cohorts (e.g., ICD-10 vs. NPIAP/UT).

Author Response: Thank you for this suggestion. After discussion, we decided to retain the current presentation with one classification tree per database. This is because there is no clear overlap between the systems for subcategories of wound types. However, we have added an explanation for this decision in the manuscript. It appears in the first paragraph under the Wound Classification section.

Comment 3: Briefly address the generalizability of findings outside Sweden and the Netherlands, especially in regions with limited digital infrastructure.

Author Response: Thank you for this comment. We agree that this is an important consideration. In the revised manuscript, we have added a reflection on the generalizability of findings to other regions, particularly those with limited digital infrastructure. This can be found in the Discussion section, in the last seven lines under the headline “Information driven care and cross-cohort analysis.”

Comment 4: Clarify how future NLP/LLM efforts might resolve missing wound ID and outcome variables in RHIP.

Author Response: Thank you for your comment. We agree that this was not sufficiently clarified in the original manuscript. In the revised version, we have added a few sentences describing why we expect NLP/LLM approaches to be helpful and how we plan to manage the inherent uncertainty. This reflection appears in the third-to-last paragraph under the “Content, quality, quantity and granularity” headline. It begins with “Discussion with clinicians have indicated...” and continues to the end of that paragraph.

We hope that these revisions adequately address the reviewers’ concerns and improve the clarity and rigor of our manuscript. We are grateful for the opportunity to revise and resubmit.

Sincerely,

The authors

---

## [Decision Letter · Decision Letter 1]

5 Dec 2025

Cohort Profile: The Dutch Wound Monitor Cohort and the Swedish Region Halland Integrated Platform (RHIP) Wound Cohort

PONE-D-25-23383R1

Dear Dr. Gustafsson,

We’re pleased to inform you that your manuscript has been judged scientifically suitable for publication and will be formally accepted for publication once it meets all outstanding technical requirements.

Kind regards,

Yih-Kuen Jan, PhD

Academic Editor

PLOS ONE

Additional Editor Comments (optional):

Reviewers' comments:

Reviewer's Responses to Questions

**Comments to the Author**

1. If the authors have adequately addressed your comments raised in a previous round of review and you feel that this manuscript is now acceptable for publication, you may indicate that here to bypass the “Comments to the Author” section, enter your conflict of interest statement in the “Confidential to Editor” section, and submit your "Accept" recommendation.

Reviewer #2: (No Response)

2. Is the manuscript technically sound, and do the data support the conclusions?

Reviewer #2: (No Response)

3. Has the statistical analysis been performed appropriately and rigorously?

Reviewer #2: (No Response)

4. Have the authors made all data underlying the findings in their manuscript fully available?

Reviewer #2: (No Response)

5. Is the manuscript presented in an intelligible fashion and written in standard English?

Reviewer #2: (No Response)

6. Review Comments to the Author

Reviewer #2: (No Response)

7. PLOS authors have the option to publish the peer review history of their article (what does this mean?). If published, this will include your full peer review and any attached files.

Reviewer #2: **Yes:** Abdulrahman Almalki

---

## [Editor Report · Acceptance letter]

PONE-D-25-23383R1

PLOS One

Dear Dr. Gustafsson,

I'm pleased to inform you that your manuscript has been deemed suitable for publication in PLOS One. Congratulations! Your manuscript is now being handed over to our production team.

Kind regards,

on behalf of

Dr. Yih-Kuen Jan

Academic Editor

PLOS One